

# The effect of varying thicknesses of mineral trioxide aggregate (MTA) and Biodentine as apical plugs on the fracture resistance of teeth with simulated open apices: a comparative *in vitro* study

Pankaj Panjwani[1], Kulvinder Banga[1], Jatin Atram[1],
Dian Agustin Wahjuningrum[2], Alexander Maniangat Luke[3,4],
Krishna Prasad Shetty[3,5] and Ajinkya M. Pawar[1]

[1] Department of Conservative Dentistry and Endodontics, Nair Hospital Dental College, Mumbai, Maharashtra, India
[2] Department of Conservative Dentistry, Faculty of Dental Medicine, Universitas Airlangga, Surabaya, Indonesia
[3] Department of Clinical Sciences, College of Dentistry, Ajman University, Ajman, United Arab Emirates
[4] Center for Medical and Bio-Allied Health Sciences Research (CMBAHSR), Ajman University, Ajman, United Arab Emirates
[5] Department of Conservative Dentistry and Endodontics, Manipal College of Dental Science MAHE, Manipal, India

Corresponding authors
Dian Agustin Wahjuningrum,
dian-agustin-w@fkg.unair.ac.id
Ajinkya M. Pawar,
ajinkya@drpawars.com

## ABSTRACT

**Background:** This study evaluates the fracture resistance of apical plugs created from Biodentine and mineral trioxide aggregate (MTA) in thicknesses of 3 and 5 mm within simulated open apex tooth models.

**Methods:** Fifty human maxillary central incisors were obtained from a pool of freshly extracted teeth. In order to replicate open apices without cavity preparation, ten teeth in the control group received apical-to-coronal preparation with Peeso reamers. The remaining 40 teeth were randomly assigned to four experimental groups and received either 3 or 5 mm Biodentine or MTA apical plugs.

**Results:** The mean fracture loads observed in this study were as follows: control group, 431.48 N (±34.55); 3 mm MTA, 774.88 N (±62.74); 5 mm MTA, 752.65 N (±73.79); 3 mm Biodentine, 918.25 N (±59.09); and 5 mm Biodentine, 903.42 N (±24.48). Specifically, teeth in the Biodentine group demonstrated considerably stronger fracture resistance compared to those in the MTA group ($p < 0.001$). However, no significant differences were observed between the 3 and 5 mm thicknesses (MTA: $p = 0.98$, Biodentine: $p = 0.99$), suggesting that plug thickness did not affect fracture resistance within both groups.

**Conclusion:** Biodentine apical plugs provided the highest fracture resistance among the materials, regardless of thickness.

## INTRODUCTION

Maxillary central incisors are among the most frequently traumatized teeth in the permanent dentition, particularly in younger populations. Traumatic dental injuries can significantly affect both oral health and aesthetics (*Wilkinson, Beeson & Kirkpatrick, 2007*). A retrospective study conducted in southern Turkey identified falls as the primary cause of dental trauma, with maxillary central incisors being the most commonly affected teeth (*Zuhal, Semra & Hüseyin, 2005*). Given that many patients avoided seeking treatment, the study emphasizes the significance of prompt dental repair following trauma. Similarly, school-age children in Yazd, Iran frequently experience severe damage to their maxillary permanent teeth, with maxillary central incisors being the most frequently injured. The most common injuries, enamel-only fractures, were found to be predominantly caused by falls, with boys being more frequently affected (*Navabazam & Farahani, 2010*). Additionally, a comprehensive study at the University of Verona, Italy investigated the distribution and frequency of severe damage to permanent incisors. The study found that children between the ages of 6 and 13 exhibited a high rate of dental trauma, with simple crown fractures resulting primarily from falls and traffic accidents being the most common injuries (*Spinas, Pipi & Dettori, 2020*).

Teeth with open apices pose significant challenges in endodontic treatment due to the lack of natural apical constriction. Traditional techniques, such as the use of calcium hydroxide for apexification, have been employed but are often time-consuming (*Gawthaman et al., 2013*). Mineral trioxide aggregate (MTA) has gained popularity as a more effective alternative that offers biocompatibility and antibacterial properties along with the ability to accelerate apical barrier formation. Regenerative endodontic therapy (RET) using scaffolds has also shown promise in stimulating tissue growth and apex closure, particularly in failed cases (*Cervino et al., 2020*). Innovative approaches such as autotransplantation with resorbable materials such as tricalcium phosphate ceramic and the use of biocompatible materials such as platelet-rich fibrin (PRF) during surgical interventions have further improved outcomes.

Single-visit apexification has become a viable treatment option for teeth with necrotic pulps and open apices. In recent years, materials such as MTA and Biodentine have been frequently used to induce apical closure, often in combination with PRF as an internal matrix (*Pham et al., 2024*). Moreover, the incorporation of laser technology into this method enhances disinfection, which may result in better treatment outcomes (*Pham et al., 2024*). This procedure is increasingly favored by patients and clinicians due to its effectiveness and practicality, reducing overall treatment duration and improving patient compliance.

Recent studies have increasingly focused on the use of MTA and Biodentine in apexification procedures for teeth with partially developed roots and open apices (*Brito-Júnior et al., 2014*; *Hachmeister et al., 2002*; *Yadav et al., 2015*). These studies primarily aim to enhance fracture resistance, improve clinical outcomes, and promote radiographic healing. Biodentine, a bioactive dentine substitute, has been developed to address many of the limitations associated with MTA. Its composition, which includes calcium silicate and

other additives, improves both its mechanical and bioactive properties, making it a suitable material for apexification.

In addition to calcium-enriched formulations, MTA and Biodentine have also shown potential as apical plugs, increasing the fracture resistance of treated teeth. Apexification therapies using MTA and Biodentine have demonstrated high clinical success rates, with some studies reporting a 100% success rate (*Puppala et al., 2021*; *Tolibah et al., 2022*; *Ravindran et al., 2022*). In particular, Biodentine has proven advantageous in revascularization due to its superior sealing ability and reduced risk of discoloration compared to MTA. This underlines the importance of understanding the unique characteristics of Biodentine and its integration into contemporary endodontic treatments.

Revascularization procedures employing MTA have shown significant improvements in root thickness, length, and overall survival rates when compared to standard calcium hydroxide apexification (*Kim et al., 2018*; *Xie et al., 2021*; *Panda et al., 2022*). Both MTA and Biodentine have demonstrated potential in strengthening immature teeth during apexification treatments (*Linsuwanont, Kulvitit & Santiwong, 2018*; *Santos et al., 2022*). Interestingly, MTA has been shown to exhibit superior fracture resistance. This is especially the case when complete obturation is performed. However, according to *Boufdil et al. (2020)*, Biodentine has demonstrated potential in one-step apexification, with follow-up studies reporting no clinical complications and satisfactory periapical tissue regeneration.

MTA remains a reliable material for orthograde filling in apexification treatments, despite Biodentine offering superior handling properties. Both materials reduce periapical radiolucency and demonstrate high healing rates when used to obturate immature teeth afflicted by periapical lesions (*Dong & Xu, 2023*; *Kamatchi et al., 2023*). MTA and Biodentine are excellent choices for apexification (*Tolibah et al., 2022*), contributing significantly to the strengthening and healing of impacted teeth while achieving favorable clinical outcomes. However, in certain cases, Biodentine might be more practical in terms of handling and therapeutic outcomes while treating teeth with open apices (*Pawar, Kokate & Shah, 2013*; *Malkondu, Kazandağ & Kazazoğlu, 2014*).

Numerous studies have evaluated the effectiveness of Biodentine and MTAs in treating immature teeth (*Brito-Júnior et al., 2014*; *Hachmeister et al., 2002*; *Yadav et al., 2015*). A study assessing the fracture resistance of immature teeth treated with MTA, Biodentine, and Bioaggregate found no significant differences among the three materials, although all considerably strengthened fracture resistance compared to untreated controls (*Bayram & Bayram, 2016*). Similarly, *Yadav et al. (2015)* demonstrated that apexification procedures in nonvital immature permanent teeth yielded a 100% clinical success rate for both MTA and Biodentine. This high success rate confirms the reliability of the materials in pediatric endodontic therapy. However, the same study found that calcium phosphate cement (CPC) showed better radiographic results after a nine-month follow-up, while MTA and Biodentine maintained similar clinical success.

Research has also advanced the understanding of the most effective methods for plug insertion and apical sealing during apexification procedures. For instance, a 4 mm apical plug is recommended to achieve effective sealing, which greatly reduces bacterial leakage

and improves the integrity of the apical seal (*Pereira et al., 2021*). Meanwhile, to promote root development and thickness while ensuring enough resistance to bacterial contamination, a material plug 5 mm short of the radiographic root end is recommended (*Torabinejad et al., 2017*). Additionally, a prospective study indicated that the apical extension of the plugs did not significantly affect treatment outcomes in young nonvital teeth (*Tabiyar & Logani, 2021*). This suggests that placement flexibility is possible without sacrificing treatment effectiveness. Nevertheless, for optimal outcomes, the combination of a 4 mm apical barrier and a material plug 5 mm short of the root end is recommended (*Tabiyar & Logani, 2021*).

The thickness of the apical plug plays a significant role in the fracture resistance of teeth during apexification. Studies have shown that in developing teeth, a 4 mm apical plug enhances fracture resistance and sealing (*Eram et al., 2020*; *Grayli et al., 2021*; *Wei et al., 2022*). Research also indicates that even plugs as thick as 6 mm can preserve the structural integrity and fracture resistance of teeth (*Mahajan, Manuja & Chaudhary, 2022*). Therefore, it is essential to take into consideration the thickness of the apical plug to achieve the best possible fracture resistance and sealing during apexification.

Several studies have recommended an ideal apical plug thickness of 3 to 5 mm (*Lawley et al., 2004*; *Bani, Sungurtekin-Ekçi & Odabaş, 2015*; *Çiçek et al., 2017*; *Dholakia & Vaidya, 2020*). However, inconsistencies remain regarding the influence of plug thickness on fracture resistance. Consequently, this *in vitro* study aims to assess the fracture resistance of simulated immature teeth treated with MTA and Biodentine apical plugs of varying thicknesses. By evaluating difference plug materials and thicknesses, this study seeks to provide clinicians with evidence-based recommendations for choosing the most suitable material and thickness for apexification treatments in immature teeth. The null hypothesis of this study proposed that there would be no significant differences in fracture resistance between teeth treated with MTA and Biodentine apical plugs and between those with 3 and 5 mm thicknesses.

## MATERIALS AND METHODS

This *in vitro* study was conducted in accordance with the principles outlined in the Declarations of Helsinki and received ethical approval from the Ethics Committee of Nair Hospital Dental College, Mumbai (EC/PG 10/CONS/2018/D-0129/2021; approval date: 28/02/2019). The Institutional Ethics Committee of Nair Hospital Dental College determined that informed consent was not necessary because this study was conducted *in vitro*. The methodology was adopted from the study conducted by *Çiçek et al. (2017)*.

### Sample size estimation

The sample size was estimated using G\*Power 3.1.9.7 based on the mean values of fracture resistance and sample sizes from previous research (*Bayram & Bayram, 2016*). An *a priori* power analysis for a one-way ANOVA (fixed effects, omnibus) using an F-test was performed to determine the required sample size. The analysis assumed an effect size (f) of 39.0099, a power (1 − β error probability) of 0.80, and an alpha error probability of 0.05.

For four groups, the analysis yielded a denominator degrees of freedom (df) of 4, a numerator degrees of freedom (df) of 3, and a crucial F-value of 6.5914. With an actual power of 1.000, the required total sample size was found to be eight samples per group, suggesting a good chance of finding an impact under the conditions being examined. However, this study included 10 samples per group to enhance robustness.

## Tooth selection

Fifty permanent maxillary central incisors, unrelated to this study, were obtained from a pool of recently extracted teeth. After removing soft tissue and calculus with ultrasonic scaling, the teeth were rinsed for an hour with 5.25% sodium hypochlorite. Under a stereomicroscope (20x Olympus SZ61 with a SC100; Richmond Hill, ON, Canada), each tooth was examined to ensure it was free of cavities, external resorption, cracks, or fractures. Buccolingual and mesiodistal periapical radiographs were taken to exclude teeth with calcifications, internal resorptions, or extra root canals. This study did not include teeth with carious lesions, calcified canals, resorptions, fractures, or cracks.

The buccolingual and mesiodistal root diameters were measured at the cervical, middle, and apical thirds using a digital Vernier caliper. To maintain uniformity throughout the investigation, only teeth with a length of $20 \pm 0.51$ mm were included. Following selection, the apical 5 mm of each tooth was cut using a carborundum disc fixed on a contra-angled handpiece.

## Treatment procedure

In the control group, teeth with immature or open apices were simulated using Peeso reamers (sizes 1–5) from the apical to the coronal direction. No access cavity preparation was carried out. This group, which included ten samples, served as a baseline for comparison with the experimental groups.

For the 40 experimental teeth, coronal access was prepared using a size-4 round bur with a high-speed handpiece. Using a barbed broach, pulps were removed manually. Peeso reamers (sizes 1–5) used to simulate immature teeth could extend 1 mm beyond the apex. Instrumentation included canal irrigation with 2.5% sodium hypochlorite. To replicate Cvek's stage 3 root growth, further canal preparation was conducted 3 mm below the cementoenamel junction using a size-6 Peeso reamer. Each canal was irrigated with 3 ml 2.5% sodium hypochlorite and 3 ml saline after instrumentation. The access cavity was temporarily sealed with Cavit (3M ESPE AG, Seefeld, Germany), and the specimens were incubated at 37 °C and 100% humidity for four weeks.

The experimental teeth were randomly divided into four groups of 10 samples each. For a 3 mm apical barrier, the material was delivered to the apical third of the canal using an MTA carrier (Oracraft, Punjab, India) and gently condensed at the apex with hand pluggers (Woodpecker Medical Instrument Co., Ltd., Xiangshan District, China), applying minimal pressure to prevent extrusion. For a 5 mm apical barrier, the same procedure was followed, with additional layers of material placed incrementally to achieve the required thickness.

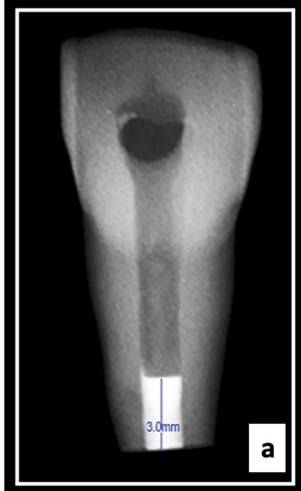
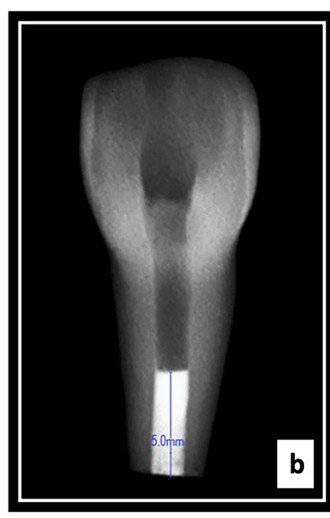
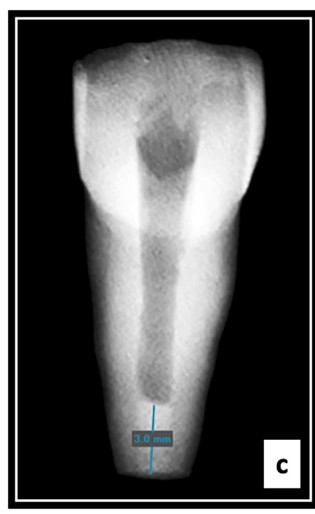
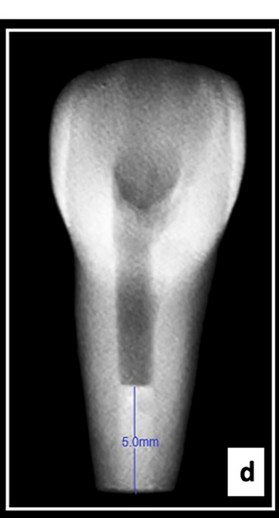

**Figure 1** Radiographs showing representative samples with apical plug thicness of the four groups tested ((A) 3 mm MTA; (B) 5 mm MTA; (C) 3 mm Biodentine; and (D) 5 mm Biodentine).

### Group I: 3 mm MTA apical plug

The simulated immature roots were orthogradely filled with MTA (ProRoot; Dentsply Sirona Endodontics, Philadelphia, PA, USA) and condensed with a hand plugger to create a 3 mm apical plug.

### Group II: 5 mm MTA apical plug

The simulated immature roots were orthogradely filled with MTA and condensed with a hand plugger to create a 5 mm apical plug.

### Group III: 3 mm Biodentine apical plug

Biodentine (Septodont Healthcare India Pvt. Ltd., Maharashtra, India) was orthogradely placed into the simulated immature roots and condensed with a hand plugger to create a 3 mm apical plug.

### Group IV: 5 mm Biodentine apical plug

The simulated immature roots were orthogradely filled with Biodentine and condensed with a hand plugger to create a 5 mm apical plug.

The homogeneity and thickness of the apical plugs were confirmed *via* radipgraphs for all the experimental groups (*Torabinejad et al., 2017*). All specimens were incubated at 37 °C and 100% humidity for four hours (Fig. 1). For all four groups, the remaining canal space up to the cementoenamel junction was obturated with gutta-percha and AH Plus sealer using the warm vertical compaction technique (Calamus Dual Obturation System Dentsply Sirona, Charlotte, NC, USA). Radiographs were taken to evaluate the quality of obturation (Fig. 2). Resin composite was used to seal the access cavities. The teeth were embedded in silicone impression material placed in a plastic container till the completion of the root canal instrumentation procedure and apical plug placement procedures. Next, for 4 weeks, all samples were incubated at 37 °C and 100% humidity.
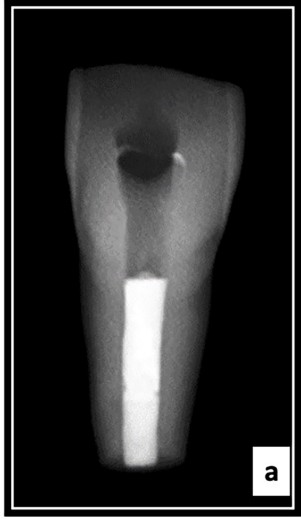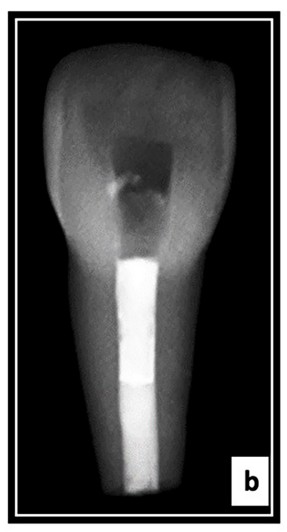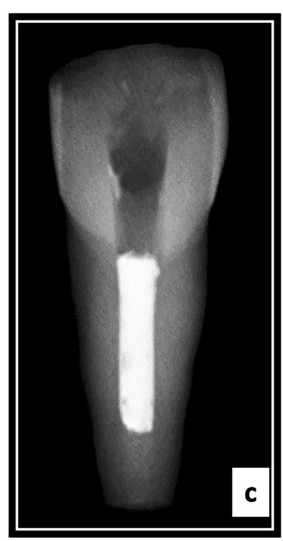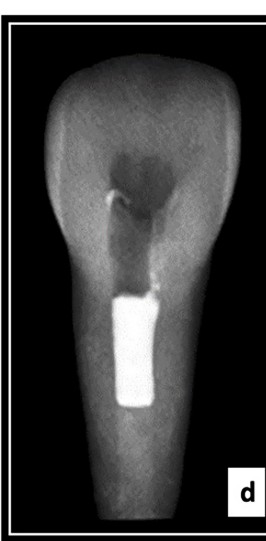

**Figure 2** Radiographs showing representative samples following obturation with GuttaPercha of the four groups tested ((A) 3 mm MTA; (B) 5 mm MTA; (C) 3 mm Biodentine; and (D) 5 mm Biodentine).

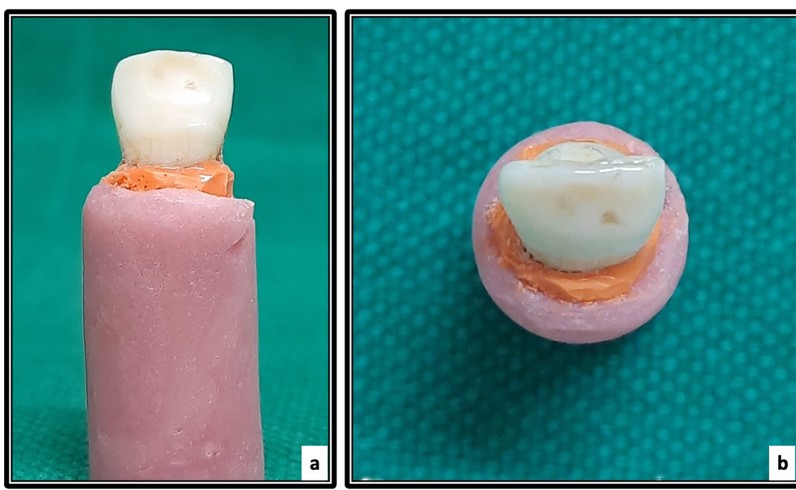

**Figure 3** Samples mounted in acrylic blocks (A) the buccal view and (B) occlusal view.

## Fracture testing

The root surfaces were coated with polyether imprint material to stimulate periodontal membrane. The roots were embedded in self-curing resin blocks, with 2 mm of the root exposed above the resin surface (Fig. 3). A universal testing machine (Instron, Norwood, MA, USA) was used to apply force. A spade was positioned buccally/lingually 3 mm above the cementoenamel junction at a 135° angle to the tooth's long axis to simulate a traumatic impact on the middle third of dental crowns. Samples were loaded at a crosshead speed of 1 mm/min until facture occurred (Fig. 4). The peak fracture load was recorded in Newtons (N).

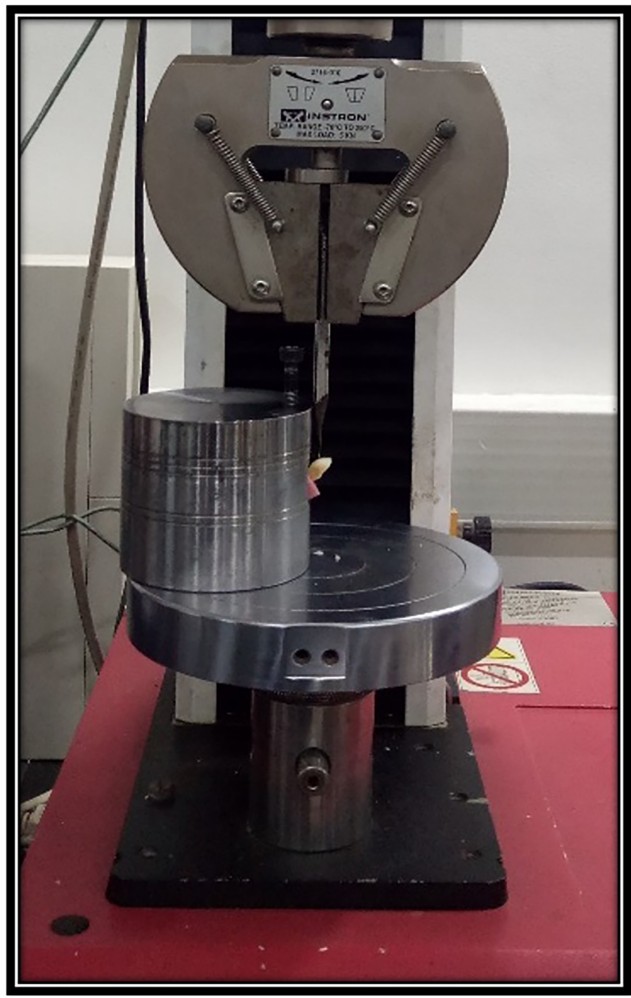

**Figure 4** Representative sample placed at 45° angle in a specially designed jig on the universal testing machine.

## Statistical analysis

Statistical analyses were performed using SPSS 16.0 (IBM, Armonk, NY, USA). The Kolmogorov-Smirnov and Shapiro-Wilk tests were used to determine the normality of the data. Differences in fracture resistance were analyzed using a one-way analysis of variance (ANOVA) followed by Tukey's *post hoc* test for multiple comparisons. A significance level of 95% confidence and a $p < 0.05$ were deemed as significant.

## RESULTS

The results of the normality test, which assessed the distribution of the data gathered for this investigation, are summarized in Table 1. The data were normally distributed, as confirmed by the Kolmogorov-Smirnov and Shapiro-Wilk tests with $p$-values greater than 0.05 across all groups.

A one-way ANOVA test was performed to examine the differences in fracture resistance among the groups. The results, which are presented in Table 1, reaffirmed the normality of

**Table 1 Test of normality: the data obtained was subjected to the Kolmogorov-Smirnov and Shapirov-Wilk tests of normalcy.**

| Groups | Kolmogorov-Smirnov[a] | | | Shapiro-Wilk | | |
|---|---|---|---|---|---|---|
| | Statistic | df | p-value | Statistic | df | p-value |
| Control | 0.17 | 10 | 0.923* | 0.93 | 10 | 0.41 |
| 3 mm MTA | 0.25 | 10 | 0.45* | 0.87 | 10 | 0.09 |
| 5 mm MTA | 0.17 | 10 | 0.92* | 0.95 | 10 | 0.47 |
| 3 mm Biodentine | 0.29 | 10 | 0.32* | 0.88 | 10 | 0.13 |
| 5 mm Biodentine | 0.13 | 10 | 0.99* | 0.95 | 10 | 0.58 |

Notes:
[a] Lilliefors significance correction.
* This is a lower bound of the true significance.

**Table 2 ANOVA test results of all the groups with inter-group comparison.**

| Groups | N | Mean | Standard deviation | Standard error | Minimum | Maximum | F value | p* value |
|---|---|---|---|---|---|---|---|---|
| Control | 10 | 431.48 | 34.55 | 48.37 | 352.50 | 475.23 | 129.86 | 0.000 |
| 3 mm MTA | 10 | 774.88 | 62.74 | 19.85 | 629.22 | 844.17 | | |
| 5 mm MTA | 10 | 752.65 | 73.79 | 23.34 | 624.45 | 861.81 | | |
| 3 mm Biodentine | 10 | 918.25 | 59.09 | 18.69 | 850.13 | 1,027.92 | | |
| 5 mm Biodentine | 10 | 903.42 | 24.48 | 7.75 | 870.43 | 956.17 | | |
| Total | 50 | 756.13 | 182.78 | 25.84 | 352.50 | 1,027.92 | | |

Note:
* One way ANOVA test.

**Table 3 Pair-wise comparison between all the groups (*post hoc* Tukey's test).**

| Pairwise comparisons | | p value |
|---|---|---|
| Control VS | 3 mm MTA | 0.000* |
| | 5 mm MTA | 0.000* |
| | 3 mm Biodentine | 0.000* |
| | 5 mm Biodentine | 0.000* |
| 3 mm MTA VS | 5 mm MTA | 0.890 |
| | 3 mm Biodentine | 0.0045* |
| | 5 mm Biodentine | 0.0136* |
| 5 mm MTA VS | 3 mm Biodentine | 0.0007* |
| | 5 mm Biodentine | 0.002* |
| 3 mm Biodentine VS | 5 mm Biodentine | 0.972 |

Note:
* *Post-hoc* Tukey's test exhibiting significant difference.

the data ($p > 0.05$). Table 2 presents the mean fracture loads for the control and experimental groups. The 3 mm Biodentine group had the highest mean fracture load, while the control group had the lowest. A significant difference in mean fracture load was observed between the control and experimental groups ($p < 0.001$).

Pairwise comparisons of the mean fracture loads between the experimental and control groups are presented in Table 3. The control group had a significantly lower mean fracture load than all experimental groups ($p < 0.01$). The groups with 3 and 5 mm MTA had comparable mean fracture loads. Similarly, no significant difference was observed between the groups with 3 and 5 mm Biodentine. However, both the 3 and 5 mm MTA groups had significantly lower mean fracture loads than the 3 and 5 mm Biodentine groups.

## DISCUSSION

Dental trauma is a serious public health concern, especially among young adults, affecting approximately 25% of children and 33% of adults (*Djemal et al., 2022*). This highlights the critical role of dental professionals in managing trauma, as young permanent teeth are particularly vulnerable to injury. Beyond physical health, dental trauma can result in pain, infection, and functional impairment, which can have an impact on general well-being. In addition, psychological effects including reduced self-esteem are frequently experienced, particularly among adolescents (*Arhakis, Athanasiadou & Vlachou, 2017*).

Endodontic therapy is crucial for salvaging traumatized teeth, yet it poses several challenges (*Nunes et al., 2019*). These include wide and divergent apical thirds, difficulties in root canal irrigation and obturation, and thin, fragile walls. Such teeth are also prone to fractures, particularly in the cervical region. Despite these advantages, calcium hydroxide apexification has several drawbacks. The treatment often requires multiple visits, prolonging therapy duration (*Murray, 2023*). Additionally, the use of intermediate restorative materials for interappointment restoration increases the risk of root canal recontamination (*Wuersching et al., 2023*). Long-term use of calcium hydroxide dressings can elevate the risk of root fracture, as its high pH of hydroxyl ions can gradually denature and disintegrate the protein structure of the dentin (*Andreasen, Farik & Munksgaard, 2002*; *Whitbeck, Quinn & Quinn, 2011*; *Batur, Erdemir & Sancakli, 2013*; *Kahler et al., 2018*). In light of these limitations, researchers and clinicians have explored alternative materials such as MTA, Bioaggregate, and Biodentine. These materials provide a more efficient approach to apexification while minimizing the risks associated with traditional calcium hydroxide treatment. This study aims to evaluate the fracture resistance of teeth with simulated open apices, treated with varying thicknesses of Biodentine and MTA apical plugs. By identifying the optimal plug thickness, this study seeks to enhance the long-term prognosis and reduce fracture risks in immature teeth undergoing endodontic treatment.

The findings of this study revealed that teeth filled with Biodentine apical plugs demonstrated significantly stronger fracture resistance than those restored with MTA apical plugs, regardless of plug thickness. Specifically, statistically significant differences ($p < 0.001$) were observed between the mean fracture loads for the Biodentine groups (918.25 N for 3 mm and 903.42 N for 5 mm) and those of the MTA groups (774.88 N for 3 mm and 752.65 N for 5 mm). This suggests that Biodentine provides superior reinforcement of immature teeth and may thus be the preferable material in apexification procedures aimed at enhancing fracture resistance.

It is interesting to note that this study found no significant difference in fracture strength between the Biodentine and MTA groups ($p = 0.99$ and $p = 0.98$, respectively) regarding the thickness of the apical plugs (3 or 5 mm). This finding contrasts with earlier research suggesting that a thicker plug improved the fracture strength of immature teeth (*Çiçek et al., 2017*; *Mahajan, Manuja & Chaudhary, 2022*; *Mohite et al., 2022*). The findings of this study indicated that a 3 mm thickness for both MTA and Biodentine may be sufficient to provide adequate mechanical strength, making clinical procedures easier while preserving structural integrity.

Apexification is a conventional treatment for immature permanent teeth with open apices that has traditionally involved the use of calcium hydroxide. Studies have documented high success rates for calcium hydroxide apexification, ranging from 74% to 96% (*Pawar et al., 2013*; *Silveira et al., 2015*; *Boufdil et al., 2020*). According to *Mohammadi, Shalavi & Yazdizadeh (2012)*, the high pH of calcium hydroxide is key to its effectiveness, which facilitated canal disinfection, reduces granulation tissue, inhibits osteoclastic activity, promotes hard tissue barrier formation, and supports periapical healing while preventing inflammatory resorption.

The significance of apical plug thickness in apexification procedures has been extensively studies. Research consistently indicates that an apical plug with a thickness between 3 and 5 mm is critical for adequate sealing, regardless of the biomaterial used (*Abbas et al., 2020*). The thickness of the apical plug plays a crucial role in enhancing the fracture resistance of immature teeth while ensuring an effective seal. In this study, a thickness of 5 mm was selected to offer an intermediate level between the widely used 3 and 6 mm plugs (*Çiçek et al., 2017*). This approach aims to provide a more versatile solution for clinical scenarios with varying canal anatomies and degrees of apical resorption.

*Çiçek et al. (2017)* found no significant difference in fracture resistance between 3 and 6 mm plugs, suggesting that a 5 mm plug could offer similar benefits across diverse clinical scenarios. Furthermore, a 5 mm plug facilitates easier handling and placement of the material during clinical procedures. This thickness provides sufficient mass to withstand fracture while allowing for effective and manageable placement of MTA without causing undue difficulties.

This finding is particularly relevant to this study as it validates the use of Biodentine and MTA as apical plug materials. Both materials improve fracture resistance, and thick apical plugs do not compromise tooth integrity. According to *Çiçek et al. (2017)*, a thickness of 4 mm enhances sealing ability and fracture resistance, supporting the idea that thicker plugs provide superior tooth protection. The findings of this study, which examined plugs with thicknesses of 3 and 5 mm, revealed no significant difference in fracture resistance.

Teeth with Biodentine apical plugs demonstrated significantly higher fracture resistant compared to those with MTA plugs. The difference could be attributed to Biodentine's enhanced resistance to dislodgement, which creates tag-like structures and a "mineral infiltration zone" upon contact with dentin (*Atmeh et al., 2012*; *Arandi & Thabet, 2021*). Furthermore, Biodentine reinforces the tooth over time by forming a thicker coating that is rich in calcium and silicate (*Han & Okiji, 2011*; *Gandolfi et al., 2013*; *Wang et al., 2023*). It

also exhibits superior dentin element absorption compared to MTA (*Han & Okiji, 2011*; *Malkondu, Kazandağ & Kazazoğlu, 2014*; *Lucas et al., 2017*). Because of its quick setting time and compositional changes, Biodentine has better handling and sealing qualities, which further improve its clinical effectiveness (*Kokate & Pawar, 2012*; *Pawar et al., 2017*; *Alazrag et al., 2020*). Biodentine's exceptional handling qualities and adaptation to cavity walls further contribute to its remarkable sealing performance. This ability to effectively seal cavity surfaces is facilitated by its tiny particle size (*Malkondu, Kazandağ & Kazazoğlu, 2014*).

Compared to MTA, Biodentine has lower porosity and pore volume, which might result in better sealing qualities (*Guerrero & Berastegui, 2018*). Additionally, its compressive strength has been observed to significantly increase over time, reaching values that are comparable to those of natural dentin (*Kaur, 2017*). A major factor in Biodentine's increased compressive strength, which supports its long-term stability and sealing effectiveness, is its low water-to-cement ratio. Together, these findings highlight the advantageous properties of Biodentine that support its improved sealing and fracture resistance, establishing it as a potential material for a variety of dental applications.

Despite these promising results, this study has several limitations. The results may not accurately reflect the intricacies of the oral environment because this study was done *in vitro*, which might affect how the performance of the materials *in vivo*. Moreover, the limited sample size constrains the generalizability of the results. Furthermore, this study focused only on two materials, potentially overlooking other relevant alternatives. Additionally, it did not evaluate the materials' long-term behavior under cyclic loading nor their long-term interactions with native tooth structures.

Future research should include larger sample sizes as well as a greater variety of materials and plug thicknesses. Long-term *in vivo* studies, particularly those evaluating the effects of cycle loading, are essential to validate the current findings and enhance their therapeutic usefulness. By addressing these limitations, future studies can expand on our findings and offer deeper insights.

## CONCLUSIONS

This study concludes that Biodentine apical plugs exhibit a higher fracture resistance than MTA as a direct consequence of Biodentine's superior handling, sealing abilities, and resistance to dislodgement. These results highlight the potential of Biodentine as an apexification material in immature permanent teeth, providing a better long-term prognosis and a reduced risk of fracture. Also, the plug thickness of either of the plugs demonstrated no significant effect on the fracture strength of the specimens.

### Funding

The authors received no funding for this work.

## Competing Interests

Ajinkya M. Pawar is an Academic Editor for PeerJ.

## Author Contributions

- Pankaj Panjwani conceived and designed the experiments, performed the experiments, prepared figures and/or tables, and approved the final draft.
- Kulvinder Banga conceived and designed the experiments, authored or reviewed drafts of the article, and approved the final draft.
- Jatin Atram conceived and designed the experiments, authored or reviewed drafts of the article, and approved the final draft.
- Dian Agustin Wahjuningrum analyzed the data, authored or reviewed drafts of the article, and approved the final draft.
- Alexander Maniangat Luke analyzed the data, authored or reviewed drafts of the article, and approved the final draft.
- Krishna Prasad Shetty analyzed the data, authored or reviewed drafts of the article, and approved the final draft.
- Ajinkya M. Pawar conceived and designed the experiments, performed the experiments, prepared figures and/or tables, and approved the final draft.

## Ethics

The following information was supplied relating to ethical approvals (*i.e.*, approving body and any reference numbers):

Permission to carry out this *in vitro* study was obtained from the Ethics Committee of Nair Hospital Dental College, Mumbai (EC/PG 10/CONS/2018/D-0129/2021; date of approval: 28/02/2019).

## Data Availability

The raw measurements are available in the Supplemental File.

## Supplemental Information

Supplemental information for this article can be found online at http://dx.doi.org/10.7717/peerj.18691#supplemental-information.

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
