# Peer review of "The effect of varying thicknesses of mineral trioxide aggregate (MTA) and Biodentine as apical plugs on the fracture resistance of teeth with simulated open apices: a comparative in vitro study"

_PeerJ, doi:10.7717/peerj.18691_

## Round 0.1 · original submission · Major Revisions

Dear authors

The manuscript requires major improvement. Please refer to the reviewers' comments which highlighted areas of concern.

Overall language, novelty, methodology including sample size.\

**Language Note:** The Academic Editor has identified that the English language must be improved. PeerJ can provide language editing services - please contact us at [email protected] for pricing (be sure to provide your manuscript number and title). Alternatively, you should make your own arrangements to improve the language quality and provide details in your response letter. – PeerJ Staff

·

Basic reporting

The article is interesting and seems to confirm the good results of new biomaterials in the treatment of teeth with open apexes. However, there is a lack of greater care with sizes and types of letters that must be standardized. Furthermore, a good English review will increase the shine of the article. There is an error in the affiliation name of one of the authors. Some words such as channels and sealants should be replaced by more appropriate terms in the English revision. I am in favor of publishing the article

Experimental design

No comment

Validity of the findings

The results are in accordance with the actual literature

Additional comments

Beware of standardized fonts

The english need to be reviewed

Reviewer 2 ·

Basic reporting

The overall language should be revised by a fluent speaker (I found minor language mistakes for example; juvenile or immature ? channel or canal ? , sealant or sealer ?).
Some sentences in the introduction section lack of suitable references (lines from 128 to 134 and line 158). Moreover, the methodology should be cited, as the presented methods is not new and it was mentioned in previous related articles.
I think article structure, figures, tables and Raw data shared are suitable.
The null-hypothesis is missing.

Experimental design

As for the originality of the presented article, in a previous published study ( https://www.ncbi.nlm.nih.gov/pmc/articles/PMC9671190/) this topic was discussed in more details. on the other hand this article lacks of many details like; MTA and Biodentine manufacturer, the application methods of these materials (as the application methods affect the sealing ability if the apical plugs https://www.mdpi.com/1660-4601/19/9/5304) and the universal tester name and brand.
Where is the gap of knowledge that was solved by this article ? what is its novelty?
The authors mentioned they collected 50 incisors for this study, how did they get this teeth ? what is the reason to collect 50 intact maxillary central incisors ???
One of the major problem in this research that the authors did not estimate the fracture mode and sites as this information is crucial in this topic.
What are the limitations of the current study?

Validity of the findings

What is the current power of this study to estimate the validity of the findings?.
In the materials and methods section, authors mentioned that they prepared 10 incisors for control group. However, in the results they mentioned they were only 4 samples ? this point is so confusing.
it is worth noting that 10 subjects in each group is underestimated in the current topic.

Additional comments

Provide a reference that determine using peeso reamers size 5 (1.5 mm) to stimulate grade 3 of cveck classification.
Was authors used calcium hydroxide in their groups? no, so what the reason behind discussing the use of this old material in the discussion section.

Reviewer 3 ·

Basic reporting

This experimental study is good and its finding will be beneficial to be used in clinical application.

However, the manuscript was ambiguous and there are rooms for improvement.

Line 1-3, it is suggested to change the title slightly to "The effects of various thicknesses of mineral trioxide aggregate (MTA) and Biodentine as apical plugs on the fracture............."

The literature review in introduction need more details and to be rewritten with corrections as suggested in my attachment. There were lack of references for most key points mentioned in introduction (line 63, 76, 87-88, 102, 115, 129-134, 138, 139.)

For improvement on introduction section, it is also suggested to highlight on MTA and Biodentine material and their uses in apexification, advantages and properties.

For better clarity, it should be send for proof reading after correction.

Experimental design

Aim of this study was stated in introduction. But the research question was slightly unclear on line 143-147. Please define it.

Method section requires certain details and explanation:

1) At line 180-182, control and tested samples had different canal preparation. Does this will cause co-founding variables to affect the result? It may need explanation on discussion section.
2) Brand and company of all materials used in this study (line 193, 199, 208).
3) Line 205- is there any other technique used in this experiment to ensure homogeneous thickness on all samples?
4) To include figure on mounted teeth samples on resin.
5) Why it was decided to choose 3mm and 5mm apical plug?. I can see study by Ersan Çiçek et al (J Endod. 2017 Oct;43(10)) used 3mm and 6mm MTA apical plug.

Validity of the findings

Tables 2 and 3 are simple for the readers to understand. But there was confusion in table 2 on number of sample of control, is that 10 or 4?

In results section, line 227-233, the description was not related to the attached table 1. Please recheck.

On discussion section, lines 249-287, these paragraphs will be better to be put under introduction if the authors decided to keep. Some repetition of literature review also noted in the discussion. Please remove.

The authors can also add the limitation or challenges faced during the study and probably write on clinical recommendation or suggestion based on the results obtained from this experiment.

Additional comments

I believe this is an original study conducted by the experts.

However, there are comments that must be clarified and corrected to ensure that manuscript is line with good findings of the experiment.

It should be suggested to do proof reading by professionals improve the manuscript language clarity.

Annotated reviews are not available for download in order to protect the identity of reviewers who chose to remain anonymous.

---

## Round 0.2 · Major Revisions

The reviewers have declined to provide a review of this revision so I have examined the manuscript myself.

1. The Discussion section must discuss the results of your study.
Sample size calculation - provide the source of reference of SD used in sample size calculation and suggestion to refer Epi info calculator (provide URL and date of access in bibliography. The formula provided may have errors.
2. Provide proof of professional English proofreading after correction. The overall manuscript still has language issues.
3. Results table - Authors need to ensure correct formatting of the table and shall not copy-paste the content of SPSS output.
4. The SD for control group was large. In such standardized experiments, if the SD was too large it indicated a problem in standardization in methodology. Is there any reason behind presenting only 4 control group in the earlier submission?
5. Scientific writing editing is required to ensure standardized bibliography style as well as in-text citation.

**Language Note:** The Academic Editor has identified that the English language must be improved. PeerJ can provide language editing services - please contact us at [email protected] for pricing (be sure to provide your manuscript number and title). Alternatively, you should make your own arrangements to improve the language quality and provide details in your response letter. – PeerJ Staff

---

## Round 0.3 · Minor Revisions

As you can see from the comments of the reviewers, there is still some language/grammar editing needed.

·

Basic reporting

I write you in regards to manuscript # PeerJ_97730 entitled " The effect of various thicknesses of Mineral Trioxide Aggregate (MTA) and Biodentine as apical plugs on the fracture resistance of teeth with simulated open apices: a comparative in vitro study".
I have revised your manuscript and the comments are listed below.
Comments:
The topic addressed in this manuscript is interesting and relevant to the field of endodontics.
Introduction
You need to correct in line 99 the space in the word “Biodentine”.
Materials & Methods
The sentence in lines 211, 212 e 213 should be placed in the Discussion section.
Although you adopted the methodology from Çiçek et al. (2017) it would be relevant:
In line 221 you should describe where and how you fix the teeth.
You should inform how you apply the apical barrier (plug) – Carrier, Manufacture, City, Country (line 231) and in the same line the hand plugger (line 239).
In line 247 the devices that you used in the warm vertical technique.
Discussion
In line 300 between “region. The purpose” you could put the paragraph line 329 until 339 ( Despite ……….. hydroxide treatment).
In lane 306 you could change “restored” to “filled”.
Conclusion
You need to conclude only the aim of the study. Then, please remove “As direct ………..to dislodgement”.
You also need to remove “It is necessary -----------advantages of Biodentine”.
Finally, in the conclusion it would be relevant emphasize the “The is no difference in apical plugs of different thicknesses.

Experimental design

Well done

Validity of the findings

High level of interest

Reviewer 3 ·

Basic reporting

Some spelling error and I've attached the pdf.

Experimental design

No comment

Validity of the findings

No comment

Additional comments

The authors have done major corrections. however some minor corrections need to be addressed.

Annotated reviews are not available for download in order to protect the identity of reviewers who chose to remain anonymous.

---

## Round 0.4 · accepted · Accept

Dear authors,

I would like to extend my congratulations on the acceptance of your manuscript, titled "The effect of varying thicknesses of mineral trioxide aggregate (MTA) and Biodentine as apical plugs on the fracture resistance of teeth with simulated open apices: a comparative in vitro study". Your work represents a valuable contribution to the field of endodontics, and we appreciate the effort and dedication you have put into this research.

We look forward to collaborating with you in the next stages of the publication process and are excited to share your findings with the wider scientific community.

Best regards,